# A Low Fidelity Virus Shows Increased Recombination during the Removal of an Alphavirus Reporter Gene

**DOI:** 10.3390/v12060660

**Published:** 2020-06-19

**Authors:** Tiffany F. Kautz, Elizabeth Jaworski, Andrew Routh, Naomi L. Forrester

**Affiliations:** 1Institute for Human Infections and Immunity, Department of Microbiology and Immunology, University of Texas Medical Branch, Galveston, TX 77555, USA; kautz@uthscsa.edu; 2Glenn Biggs Institute for Alzheimer’s & Neurodegenerative Diseases, University of Texas Health San Antonio, San Antonio, TX 78229, USA; 3Department of Biochemistry and Molecular Biology, University of Medical Branch, Galveston, TX 77555-0645, USA; eljawors@utmb.edu (E.J.); alrouth@utmb.edu (A.R.); 4Institute for Human Infections and Immunity, Department of Pathology, University of Texas Medical Branch, Galveston, TX 77555, USA; 5School of Life Sciences, University of Keele, Keele ST5 5BG, UK

**Keywords:** arbovirus, alphavirus, recombination, sequencing, fidelity

## Abstract

Reporter genes for RNA viruses are well-known to be unstable due to putative RNA recombination events that excise inserted nucleic acids. RNA recombination has been demonstrated to be co-regulated with replication fidelity in alphaviruses, but it is unknown how recombination events at the minority variant level act, which is important for vaccine and trans-gene delivery design. Therefore, we sought to characterize the removal of a reporter gene by a low-fidelity alphavirus mutant over multiple replication cycles. To examine this, GFP was inserted into TC-83, a live-attenuated vaccine for the alphavirus Venezuelan equine encephalitis virus, as well as a low-fidelity variant of TC-83, and passaged until fluorescence was no longer observed. Short-read RNA sequencing using ClickSeq was performed to determine which regions of the viral genome underwent recombination and how this changed over multiple replication cycles. A rapid removal of the GFP gene was observed, where minority variants in the virus population accumulated small deletions that increased in size over the course of passaging. Eventually, these small deletions merged to fully remove the GFP gene. The removal was significantly enhanced during the passaging of low-fidelity TC-83, suggesting that increased levels of recombination are a defining characteristic of this mutant.

## 1. Introduction

RNA viruses are notoriously error-prone due to the lack of an RNA-dependent RNA polymerase (RdRp) proofreading enzyme (with the unique exception of proofreading exonucleases present in coronavirus genomes). This error results in approximately one mutation per round of genome replication [1], which due to the rapid replication of RNA viruses, culminates in a broad cloud of related virus genomes, commonly referred to as viral intra-host diversity, and in specific cases as a quasispecies. If the intra-host viral population is either too diverse or too clonal, this can impact the fitness of the virus population, which commonly results in a reduction in host tropism or virulence [2,3,4,5]. While the quasispecies theory has been studied extensively in silico, in vitro, and in vivo by analyzing the ratios of nucleotides at specific genomic sites [5,6,7], the contribution of recombination between RNA virus minority variants and the effects on genome maintenance and evolution have been comparatively understudied. 

RNA virus recombination primarily occurs when the RdRp switches from the template RNA strand to an acceptor RNA strand (i.e., copy choice recombination) or, less frequently, by ligation of cleaved RNA strands [8]. Both homologous recombination, where the RdRp switches to a template with high sequence similarity, and nonhomologous recombination, where the RdRp switches to a template with low sequence similarity, can occur [9]. The molecular determinants of RNA recombination are not well-defined, but RNA recombination has been described for viruses such as poliovirus and brome-mosaic virus to arise more frequently in regions with high amounts of RNA secondary structure and AU-rich sequences [9,10]. Additionally, other factors, such as sequence homology between RNA strands and altered replication kinetics have been implicated in changing the rate of RNA recombination [9]. Rates of RNA recombination vary between different families of RNA viruses. For example, Picornaviruses have relatively high rates of recombination [11] compared to families such as Flaviviridae where viable recombination is less common [12]. 

Venezuelan equine encephalitis virus (VEEV) is an alphavirus from the family Togaviridae. Alphavirus genomes are composed of a positive-sense, monopartite single-stranded RNA that is approximately 11.5 kb in length. The first two-thirds of the genome encodes the nonstructural proteins, while the last third of the genome encodes the structural proteins under a subgenomic promoter [13]. Inter-strain of inter-species RNA recombination is not thought to be common for members of the alphavirus genus due to superinfection exclusion [14,15,16], which occurs rapidly upon cellular infection and precludes the high multiplicity infections believed to be required for frequent recombination events between co-infecting viral genomes [17]. Nonetheless, alphavirus coinfection has been described [18], and it is thought that the recombination of Sindbis and eastern equine encephalitis virus-like ancestors, two different alphaviruses, produced western equine encephalitis virus [19,20]. Conversely, the formation of defective viral genomes (DVGs) of alphaviruses has been well described for numerous family members. DVGs are generated due to nonhomologous RNA recombination events either intra-molecularly or between two copies of a viral genome resulting in shorter recombined versions of the parental genome missing critical regions required for autonomous replication. We have extensively characterized RNA recombination in the alphavirus, chikungunya virus, and demonstrated that RNA recombination events are plentiful even under low MOI and low passage conditions resulting in the synthesis of DVGs de novo after viral infection [21]. 

Beyond the role that RNA recombination plays during the natural replication-cycle of RNA viruses both in vitro and in vivo, it is important to characterize the role of RNA recombination in the stability of recombinantly engineered strains of RNA virus containing inserted reporter genes (e.g., green fluorescent protein (GFP), luciferase). Increasing the understanding of where these recombination events occur may allow for a more rational design of how to insert foreign genetic material into viral genomes to increase their genetic stability, as well as how to reduce the impact that these insertions commonly have on virus fitness. Here, we inserted GFP into a live-attenuated vaccine for Venezuelan equine encephalitis virus, TC-83, and passaged the virus using a physiologically relevant low multiplicity of infection (MOI). Illumina sequencing using ClickSeq library preparation [22] was used to determine minority variant recombination events in the genome. We found a slow but persistent removal of the GFP gene that was highly specific to the inserted gene. Additionally, we found that a low-fidelity variant of TC-83 [23] removed the GFP reporter gene much more efficiently.

## 2. Methods

### 2.1. Cell Culture and Viruses

Vero (African green monkey kidney) cells were obtained from the American Type Cell Collection (ATCC, Bethesda, MD, USA) and maintained in DMEM (Gibco, Gaithersburg, MD, USA) supplemented with 10% Fetal Bovine serum (Atlanta Biologicals, Flowery Branch, GA, USA), and 500 μg/mL gentamycin (Corning, Corning, NY, USA) in a 37 °C 5% CO_2_ incubator. A TC-83 GFP infectious clone was kindly provided by Dr Scott Weaver (UTMB), and standard cloning protocols were used to insert the GFP gene into the low-fidelity TC-83 genome. As described previously [23], low-fidelity TC-83 contains three coding mutations to the RdRp, which were found to lower the mutation frequency produced by this mutant during replication. Following this, the plasmids were transcribed, and the virus was rescued by electroporation (EP) using BHK cells. Titers of the EP stock virus were determined by the plaque assay using Vero cells [24].

### 2.2. TC-83 In Vitro Passaging: GFP

One day prior to infection, a six-well plate was seeded with 500,000 Vero cells. The day of infection, the TC-83 GFP or TC-83 3x GFP (low-fidelity TC-83) virus was diluted 10,000-fold for an MOI of approximately 0.5. Once diluted, media was decanted from the six-well Vero cell plate, and 200 μL was used to infect each well. These infections were done using three replicates per virus. Following infection, the plate was transferred to a 37 °C 5% CO_2_ incubator for 1 h with occasional rocking. When the incubation was complete, the inoculum was removed, and 2mL of DMEM (Gibco, Gaithersburg, MD, USA) supplemented with 2% FBS (Atlanta Biologicals, Flowery Branch, GA, USA) and 500 μg/mL gentamycin (Corning, Corning, NY, USA) was added to each well. The plates were then incubated for 24 h in a 37 °C 5% CO_2_ incubator. Following this, the cell supernatant was removed, transferred to a tube with FBS (Atlanta Biologicals, Flowery Branch, GA, USA) to a final concentration of 20% FBS, and centrifuged at for 5 min at 0.8 rcf to remove debris. Virus aliquots were stored at −80 °C. Plaque assays were used to determine virus titer [24]. Ten total passages were performed for each virus and replicate.

To determine PFU:GFP, plaque assays were performed as per the usual protocol [24]. After 48 h, the cells were visualized under a fluorescent microscope, and fluorescent plaques were counted. Following this, the cells were fixed and stained following the standard plaque assay protocol, and the number of plaques was counted. The number of GFP to plaque forming units (PFU) was then calculated as a ratio (i.e., # plaques/# GFP fluorescent foci) to determine the amount of GFP fluorescence lost over the course of passaging.

### 2.3. RNA Sequencing: ClickSeq Virus Recombination Analysis

Viral RNA was extracted using the Zymo Direct-zol RNA mini kit as per the manufacturer’s protocol (Zymo, Irvine, CA, USA). RNA sequencing libraries were made following the standard ClickSeq protocol using 100 ng of viral RNA, as previously described [22,25]. ClickSeq was used instead of the traditional Illumina library procedures to avoid the errant recombination that frequently occurs during the typical fragmentation and ligation steps. cDNA libraries between 400–700 bp were excised and purified using the Zymo Research Gel Recovery kit (Zymo, Irvine, CA, USA). Sequencing was performed by the University of Texas Medical Branch next-generation sequencing core using an Illumina NextSeq 550 obtaining 1 × 150 bp reads.

Cutadapt [26] was used to remove adaptor sequences. Following this, the FASTX toolkit [27] was used to remove the random hexamer sequences generated during cDNA synthesis, as well as sequences with a PHRED score below 20. Bowtie [28] was used to align the virus sequences to the reference genome (Accession number, L01443), allowing for one mismatch. Those sequences that did not align using Bowtie were processed using ViReMa [29], a viral recombination mapper, to determine the incidence and location of recombination events. These reads were normalized to the average number of reads aligning to the virus genome. M-fold [30] was used from nucleotide position 6501 to the end of the virus genome to determine the secondary structure of the virus genomes and how this related to the observed recombination events.

Raw FASTQ data are available on the NCBI Small Read Archive under BioProject ID PRJNA634635.

## 3. Results

### 3.1. Rapid Loss of GFP Fluorescence Occurred during Low-Fidelity TC-83 Passaging

To examine reporter gene stability, three independent replicates of TC-83 GFP and low-fidelity TC-83 GFP were passaged 10 times on Vero cells using an approximate MOI of 0.5 (Figure 1A). Over the course of this passaging, reduced GFP expression for TC-83 was observed beginning at passage 8 (Figure 1B). In contrast, GFP loss was observed much earlier for the low-fidelity TC-83 replicates, typically by passage 4 (Figure 1C). 

To quantify this loss, the number of fluorescent plaques was compared to the number of overall plaques for passages 5–10. Until passage 10, the majority of TC-83 plaques were fluorescent, and the largest PFU:GFP difference was approximately 1:30, which was only observed for one replicate during the final passage (Figure 1D). In contrast to this, all low-fidelity TC-83 replicates experienced an approximately 500–1500-fold decrease in fluorescence by passage 10, which was orders of magnitude higher than the ratios observed for the TC-83 GFP passage replicates (Figure 1E). As a low-fidelity Sindbis virus mutant was previously found to produce defective interfering (DI) particles at high rates [31,32], we hypothesized that the low-fidelity TC-83 was also recombining at higher rates than TC-83, resulting in rapid GFP gene removal. 

### 3.2. Low-Fidelity TC-83 GFP Gene Removal Is Enhanced

ClickSeq was used to generate Illumina RNA sequencing libraries, allowing for the determination of recombination hotspots as well as the frequency of virus deletion mutants. All three TC-83 GFP replicates contained deletion variants present at a low frequency that were able to selectively remove the GFP reporter gene by passage 10 (Figure 2A). The deletion inside the GFP gene was higher than for any other gene and occurred as soon as passage 1. In support of our hypothesis, low-fidelity TC-83 GFP was much more efficient at removing the GFP gene, as the frequency of deletions within the GFP gene was far higher than those of TC-83 GFP, which was consistent among all replicates (Figure 2B). 

The first replicate of each virus was chosen for further analysis, as these both had median levels of decreased GFP fluorescence in Figure 1D,E. For both viruses, peak recombination first appeared in the middle of the GFP gene, and progressively removed additional nucleotides over the course of passaging until the entire GFP gene was removed (Figure 3). Low-fidelity TC-83 was able to efficiently remove the entire GFP fragment by passage 3, while TC-83 required five additional passages to reach similar levels of GFP removal. This clearly demonstrates that low-fidelity TC-83 GFP produces recombination variants more frequently than TC-83 GFP.

### 3.3. Low-Fidelity TC-83 Exhibits Conserved Minority Variant Removal of the GFP Gene

To determine if the regions where recombination occurred most frequently were conserved between replicates, the top 5 recombination events for each virus and replicate were identified (Appendix A). With few exceptions, all of the most common recombination events were deletions found within and around the GFP gene, regardless of the virus backbone.

Of the top five recombination events for each passage and replicate, all the replicates of low-fidelity TC-83 predominately exhibited a deletion from nucleotide positions 7899–7990 for the first five passages. This happened concurrently with another deletion event, 7894–7985, which occurred slightly less frequently than the 7899–7990 deletion. Following the appearance of these two deletions, a much larger removal occurred from nucleotides 7894–8421, which encompassed the entire GFP gene (Figure 4B). In one replicate, this deletion was found to be one of the most predominant deletions after only one passage, and in another replicate, this became the predominate deletion by passage 2. By passage 5, this 7894–8421 deletion quickly became the most common deletion event by 1–2 orders of magnitude for all low-fidelity TC-83 GFP replicates.

Unlike the low-fidelity mutant, TC-83 did not show the same conservation of deletions between replicates and passages, but, like low-fidelity TC-83, the 7899–7990 deletion was one of the most frequently observed. The other two deletions observed during low-fidelity TC-83 passaging were also sporadically found as one of the five most common recombination events, but not consistently between replicates. However, while not always the most common of mutations, these three mutations still occurred during TC-83 GFP passaging and at levels comparable to the low-fidelity TC-83 virus (Figure 4A). Conversely, the level of the variance observed between the replicates of TC-83 GFP and low-fidelity TC-83 GFP was large, with higher variances observed between the replicates for TC-83 GFP than for low-fidelity TC-83 GFP. Other than these three deletions, all TC-83 GFP replicates produced the majority of the deletions surrounding nucleotides 7559–7574. The length and frequency of these deletions were variable but suggested that this region of the GFP gene was the most amenable to deletions and insertions of various forms.

To better understand the mechanisms underlying the conserved removal of the low-fidelity TC-83 GFP gene, M-fold was used to determine the RNA secondary structure of the virus genome (Figure 4C–E). The 7894–7895 and 7899–7990 deletions were found close together in the primary RNA sequence, and the sequence found between 7894–7899 was identical to the sequence found between 7895–7990, which was indicative of homologous recombination. This five base pair sequence (GGCAA) was similarly structured between 7894–7899 and 7985–7990, with three G-C nucleotides forming a small stem that led into a loop. As previously stated, once these two deletions occurred, a 7562–8421 deletion surrounding the GFP gene appeared and rose to prominence. Like the other two deletions, a nearly identical sequence was found at the beginning and end of the deletion (CUAGA vs. CUAAGA), showing micro-homology at the site of the recombination juncture. Unlike the previous deletion, these sequences were found in a loop structure and were far away from each other in the predicted secondary structure.

## 4. Discussion

It is well known that the virus reporter genes (e.g., GFP, luciferase) commonly used to answer questions related to virus lifecycles are unstable, with papers commonly reporting a 10–50% loss of fluorescence upon one virus passage [33,34,35,36,37]. However, it is not well understood how reporter gene removal occurs in virus populations because previous research has typically been limited to the use of low-throughput sequencing methods, generally by Sanger sequencing of a handful of virus clones surrounding a small area of the virus genome. RNAseq library synthesis methods are often prone to the generation of chimeric short-reads, giving rise to artifactual recombination in the output data obfuscating the detailed analysis of rare and low-frequency RNA recombination [22]. However, the ClickSeq method for RNAseq library preparation is specifically designed to reduce this artifact, thus allowing for the robust analysis of minority recombination events across the entire virus genomes.

In this paper, we used ClickSeq to better understand how GFP is removed when inserted into an alphavirus genome under a double subgenomic promoter. This technique allowed us to visualize the thousands of recombination events that occurred during virus passaging, which were predominantly found within the inserted GFP reporter gene. We originally hypothesized that GFP removal would primarily occur across the homologous promoter sequences surrounding the GFP gene. However, while true in later passages, the first GFP deletion events actually occurred in the middle of the GFP gene via the removal of smaller segments, and thus inactivating GPF expression before spreading outwards towards the promoter sequences. This removal was enhanced in a low-fidelity TC-83 variant.

Perhaps counterintuitively, the specific nucleotide deletions within the GFP gene were more conserved among the low-fidelity TC-83 GFP replicates than the wild type virus. This is similar to previous research with a low-fidelity Sindbis virus where a conserved deletion mutant was found at elevated levels during its passaging at a high MOI of 25 [31]. The wild type Sindbis virus demonstrated no such conservation under similar passaging conditions. Additionally, a high-fidelity variant of poliovirus appeared to only use one type of deletion to ablate the utility of a detrimental inserted miRNA, while the wild type virus displayed little conservation in the miRNA sequence deletion [38]. Together, these results suggest that high and low-fidelity RNA virus mutants are more predictable in their methods of genome pruning than their wild type counterparts, which is ideal for the use of these mutants during live-attenuated vaccine and transgene delivery design. These results also suggest that altered recombination potential may be common among fidelity variants. This is unsurprising as altering fidelity also changes the speed and processivity of the RdRp [39,40,41,42,43,44,45], which has been linked to changes in polymerase stuttering and recombination frequency [46]. However, there is evidence that at least some fidelity mutants do not exhibit large changes in recombination frequency or vice-versa [32,47], although this is important to reexamine using updated techniques. In the future, using ClickSeq to analyze virus sequence diversity and recombination would allow both of these aspects to be examined using the same dataset, which will clarify the link between replication fidelity and recombination.

The mechanism of GFP removal during low-fidelity TC-83 GFP passaging primarily occurred through the formation of three deletions. The first two deletions, Δ7894–7895 and Δ7899–7990, are found within the GFP gene and were in the highest quantities for all three replicates early during passaging. The other deletion, Δ7562–8421, fully encompasses GFP and the surrounding promoters and occurred with reduced frequency until later passages. While these deletions were also found during TC-83 GFP passaging, their fixation in the population was obscured by other deletions. By later TC-83 GFP passages, almost all of the most common deletions were found between nucleotides 7559–7574, which are within the GFP subgenomic promoter. While the most frequent deletions for TC-83 GFP and low-fidelity TC-83 GFP were different, our results show that the most common deletions were not usually due to sequence homology between the two identical promotors surrounding the GFP gene but were instead due to multiple small deletions.

To characterize the role of RNA structure of the virus genome in RNA recombination site selection, we modeled the three most common deletions that occurred at high levels during low-fidelity TC-83 GFP passaging. The first two deletions, Δ7894–7895 and Δ7899–7990, were close together in the primary sequence and shared an identical sequence and structure at the 5′ and 3′ ends of their shared deletion space in a clear act of homologous recombination. In the future, it would be interesting to see if disruption of the homologous recombination sites shared by Δ7894–7895 and Δ7899–7990 could help stabilize the GFP gene, as other deletions appeared to require more replication cycles to accumulate. The larger recombination event, Δ7562–8421, demonstrated microhomology, with nearly identical sequences around the 5′ and 3′ end of the deletion.

In our study, we only examined the stability of the GFP gene when placed between two identical subgenomic promoters. In the future it would be interesting to determine how placement of the GFP gene within the virus genome could change the most common deletions within and around GFP. Additionally, it would be worthwhile to examine other fidelity variants using ClickSeq to determine whether our observations were conserved among other fidelity mutants. However, here we demonstrate that deletions within the GFP gene are initially more common than deletions spanning the entire GFP gene, even when the gene is surrounded by identical promoter sequences. Additionally, some of these deletions are common between replicates, which could be used in the future to change the secondary structure of the GFP gene while maintaining the protein sequence to aid in increasing the stability of the gene. In summary, our research has provided clear evidence for the utility of ClickSeq both translationally for use in understanding reporter gene stability as well as enhancing our understanding of the changes in recombination that occur during low-fidelity virus replication.

## Figures and Tables

**Figure 1 viruses-12-00660-f001:**
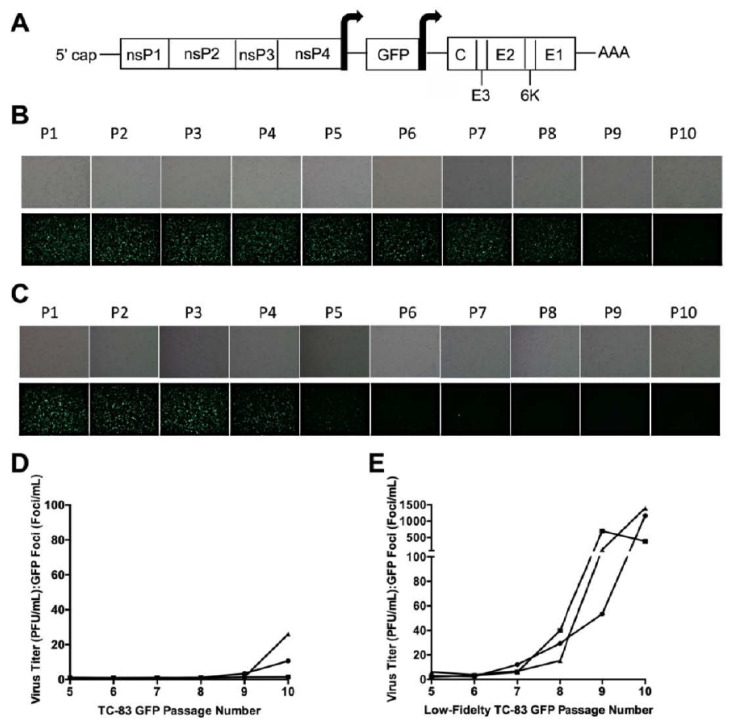
Loss of GFP fluorescence during passage. Genome schematic of TC-83 GFP (**A**). The arrows surrounding the GFP gene represent the subgenomic promotor sequences. Vero cells were infected with TC-83 GFP (**B**) or low-fidelity TC-83 GFP (**C**) over the course of 10 virus passages. The first replicate for each virus is depicted. Upper panels show brightfield microscopy images, while lower panels show GFP fluorescence. P: passage. The ratio of the PFU:GFP foci was determined for passages 5–10 to estimate the loss of GFP fluorescence for TC-83 GFP (**D**) and the low-fidelity TC-83 GFP (**E**). Replicate 1: circle, replicate 2: square, replicate 3: triangle.

**Figure 2 viruses-12-00660-f002:**
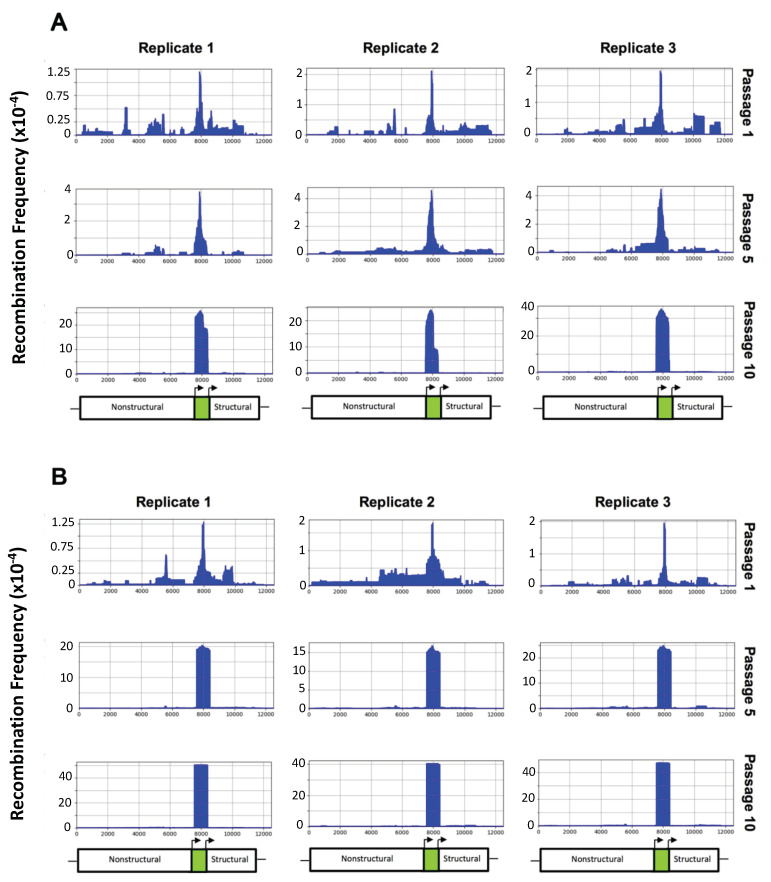
ViReMa recombination results for TC-83 GFP and low-fidelity TC-83 GFP. Passages 1, 5, and 10 are shown for each TC-83 GFP (**A**) and low-fidelity TC-83 GFP (**B**) replicate. The *x*-axis shows genome position, while the *y*-axis shows the recombination frequency (i.e., normalized to the total reads that map to TC-83 GFP). Below the recombination charts, a schematic shows the approximate location of the nonstructural protein, GFP, and structural genes.

**Figure 3 viruses-12-00660-f003:**
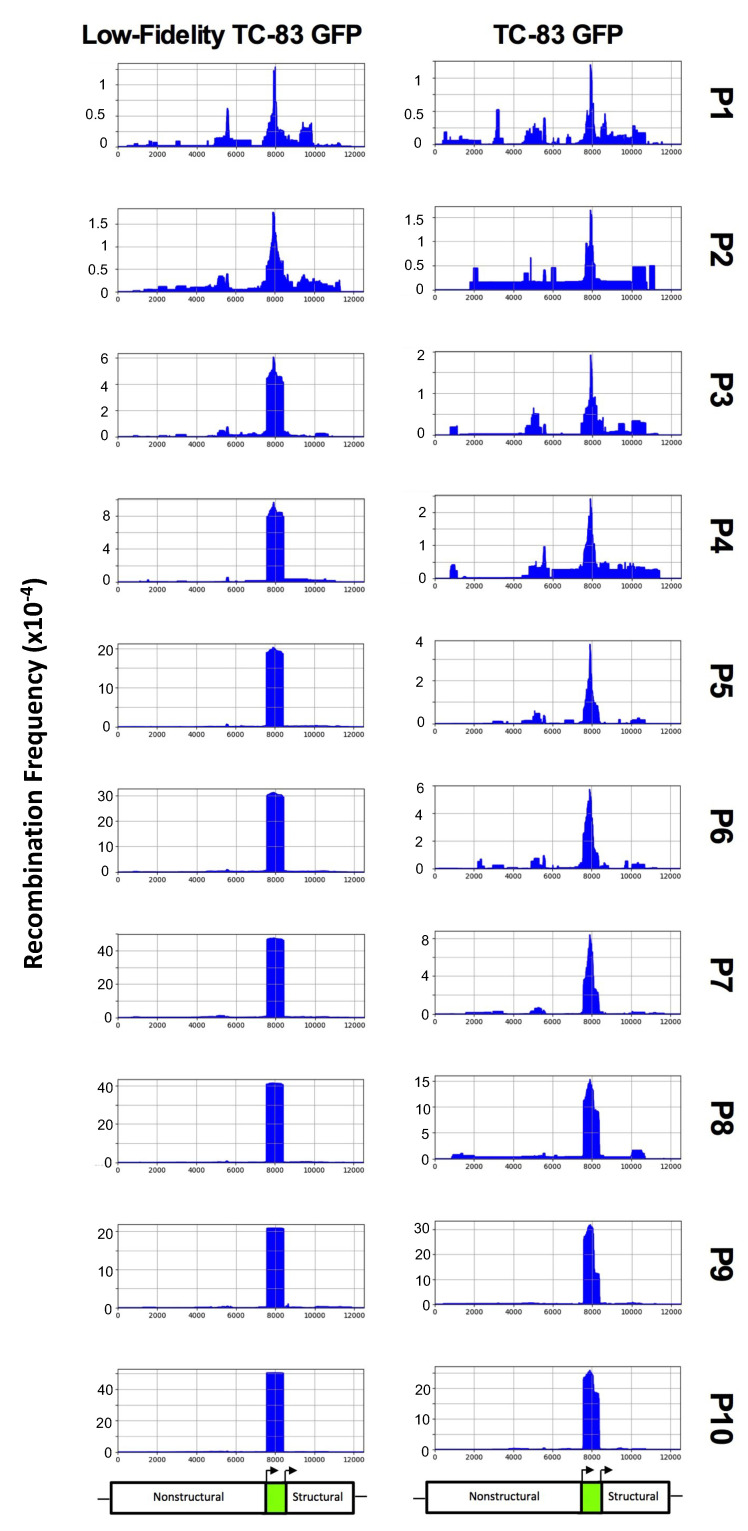
ViReMa recombination results for TC-83 GFP and low-fidelity TC-83 GFP. Each passage of the first replicate of TC-83 GFP and low-fidelity TC-83 GFP was sequenced using ClickSeq libraries and analyzed using ViReMa. The *x*-axis shows genome position, while the *y*-axis shows the recombination frequency (i.e., normalized to the total reads that map to low-fidelity TC-83 GFP). Below the recombination charts, a schematic shows the approximate location of the nonstructural protein, GFP, and structural genes. “P” represents passage.

**Figure 4 viruses-12-00660-f004:**
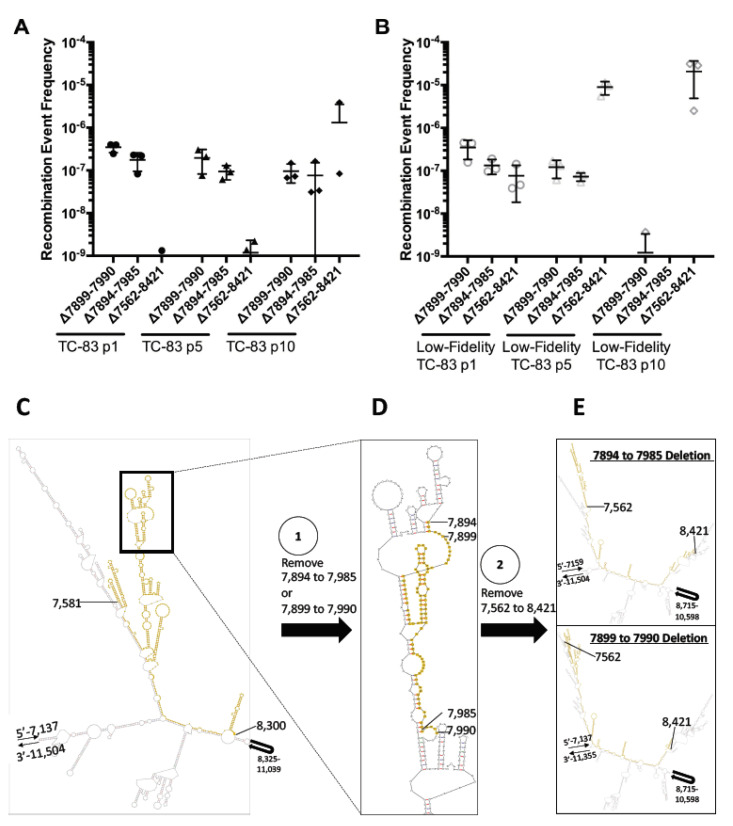
Frequency of the three most common deletion events during TC-83 GFP and low-fidelity TC-83 GFP passaging and RNA structure locations of these events in low-fidelity TC-83 GFP. As TC-83 GFP (**A**) and low-fidelity TC-83 GFP (**B**) were passaged, three deletions (Δ7894–7895, Δ7899–7990, and Δ7562–8421) rose to prominence, especially in the low-fidelity passages. Frequency of the recombination events was determined by dividing the number of recombination events by the total number of sequencing reads. If one of the three deletion events depicted above was not found for a passage replicate (i.e., below the limit of detection), no data point was depicted. The RNA structure of low-fidelity TC-83 was determined using M-fold (ΔG = −1757.85) (**C**–**E**). The GFP gene (nucleotides 7581–8300) is highlighted in yellow (**C**). During passages 1–5, the nucleotide deletions 7894–7985 and 7899–7990 (yellow highlight) concurrently arose in all replicates (**D**). An identical nucleotide sequence, GGCAA, was found at the beginning and end of these initial deletions. After these deletions, a 7562–8421 nucleotide deletion (yellow highlight) arose and became the dominant deletion mutant in the population, resulting in the total excision of the GFP gene (**E**). A nearly identical sequence was found at the beginning and end of this deletion juncture (CUAGA vs. CUAAGA).

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
