# Peer review of "A Low Fidelity Virus Shows Increased Recombination during the Removal of an Alphavirus Reporter Gene"

_viruses, 2020, doi:10.3390/v12060660_

Round 1
Reviewer 1 Report
In this paper, the authors characterized the minor variant-level recombination of a live-attenuated alphavirus and its low-fidelity variant by short-read RNA sequencing using ClickSeq. They found a slow but persistent removal of the GFP gene that was highly specific to the inserted gene. They also revealed that a low-fidelity variant of this virus removed the GFP reporter gene much more efficiently.
Techniques used in this paper, ClickSeq, are reliable and effective to revealing minor recombination events. Experiments in this study seem to have been competently conducted. Moreover, the results are clearly presented and interesting. This report represents a valid contribution to the field. Only one specific comment is below.
Figure4: It is difficult for readers to see the secondary structure in Fig. 4E. Also, it will be informative if the authors present the Gibbs free energy of the predicted structure.
Author Response
Figure4: It is difficult for readers to see the secondary structure in Fig. 4E. Also, it will be informative if the authors present the Gibbs free energy of the predicted structure.
Response: The picture quality was improved for this figure and all other figures. Gibbs free energy was also added to Fig 4 (line 287).
Reviewer 2 Report
This was a very interesting paper to read and I enjoyed it. I have some minor comments that I would like them to be address. I think they will improve the paper:
1. It is not clear what is different between TC-83 and low-fidelity TC-83, please add a detailed explanation, this will help the reader.
2. Improve de contrast in figures 1B and 1C, for both types of micrographs. It is very hard to say anything. Perhaps changing the arrangement will help; figure 1 could be divided into two figures.
3. Why si I there is diversity in the recombination sites in P1 and P2 much higher? It is not clear to me why in the first couple of passages there is recombination events everywhere in the genome (although is clear that is much higher for GFP than for the rest of the genome) but then they all reduce to be only at the reporter gene. I think this explanation will help the reader to appreciate and understand the results.
4. Line 266, Do they mean low-fidelity instead of high-fidelity? If not, I find this explanation very confusing, please re-write it to make it clearer
5. Could the recombination events be an artifact of adding a second subgenomic promoter? Would they expect this to happen if GFP were to be fused to a viral protein (with a cleavage site like FMDV between both proteins)? That is, if the reporter gene were to be flanked by coding viral sequences would the recombination rate be altered as when the reporter gene is added between subgeneric promoters?
Author Response
1. It is not clear what is different between TC-83 and low-fidelity TC-83, please add a detailed explanation, this will help the reader.
Response: Additional information about this mutant was added from lines 93-95.
2. Improve de contrast in figures 1B and 1C, for both types of micrographs. It is very hard to say anything. Perhaps changing the arrangement will help; figure 1 could be divided into two figures.
Response: The picture quality was improved for this figure and all other figures.
3. Why is I there is diversity in the recombination sites in P1 and P2 much higher? It is not clear to me why in the first couple of passages there is recombination events everywhere in the genome (although is clear that is much higher for GFP than for the rest of the genome) but then they all reduce to be only at the reporter gene. I think this explanation will help the reader to appreciate and understand the results.
Response: think this is comment is due to the axes font being small and difficult to read. Additionally, the y-axis are much lower for the first passage than they are in subsequent passages. The text size for these figures has been increased to aid the reader.
4. Line 266, Do they mean low-fidelity instead of high-fidelity? If not, I find this explanation very confusing, please re-write it to make it clearer
Response: To make this less confusing, the discussion of the low-fidelity Sindbis virus deletion conservation was changed to be discussed before the high-fidelity poliovirus deletion conservation (lines 316-318). We find it very interesting that both low and high fidelity viruses appear to be more predictable in how they change their virus sequences than viruses with wild-type replication machinery.
5. Could the recombination events be an artifact of adding a second subgenomic promoter? Would they expect this to happen if GFP were to be fused to a viral protein (with a cleavage site like FMDV between both proteins)? That is, if the reporter gene were to be flanked by coding viral sequences would the recombination rate be altered as when the reporter gene is added between subgeneric promoters?
Response: We briefly discuss this from lines 361-363 as future directions. It will be very interesting to explore this arena in the future.